# Associations of Sociodemographic Characteristics with Food Choice Motives’ Importance Among Mexican Adults: A Cross-Sectional Analysis

**DOI:** 10.3390/foods14020158

**Published:** 2025-01-07

**Authors:** Miguel Amaury Salas-García, María Fernanda Bernal-Orozco, Andrés Díaz-López, Alejandra Betancourt-Núñez, Pablo Alejandro Nava-Amante, Barbara Vizmanos

**Affiliations:** 1Departamento de Clínicas de la Reproducción Humana, Crecimiento y Desarrollo Infantil, División de Disciplinas Clínicas, Centro Universitario de Ciencias de la Salud (CUCS), Universidad de Guadalajara (UdeG), Guadalajara 44340, Mexico; amaury.salas@alumnos.udg.mx (M.A.S.-G.); alejandra.bnunez@academicos.udg.mx (A.B.-N.); pablo.nava@alumnos.udg.mx (P.A.N.-A.); 2Laboratorio de Evaluación del Estado Nutricio, Departamento de Clínicas de la Reproducción Humana, Crecimiento y Desarrollo Infantil, División de Disciplinas Clínicas, Centro Universitario de Ciencias de la Salud (CUCS), Universidad de Guadalajara (UdeG), Guadalajara 44340, Mexico; 3Instituto de Nutrigenética y Nutrigenómica Traslacional, Departamento de Biología Molecular y Genómica, División de Disciplinas Básicas, Centro Universitario de Ciencias de la Salud (CUCS), Universidad de Guadalajara (UdeG), Guadalajara 44340, Mexico; 4Departamento de Salud Pública, División de Disciplinas Para el Desarrollo, Promoción y Preservación de la Salud, Centro Universitario de Ciencias de la Salud (CUCS), Universidad de Guadalajara (UdeG), Guadalajara 44340, Mexico; 5Serra Hunter Fellow, Universitat Rovira i Virgili (URV), 43204 Reus, Spain; andres.diaz@urv.cat; 6Nutrition and Mental Health Research Group (NUTRISAM), Universitat Rovira i Virgili (URV), 43204 Reus, Spain; 7Institut d’Investigació Sanitària Pere Virgili (IISPV), 43204 Reus, Spain; 8CIBERobn Physiopathology of Obesity and Nutrition, Institute of Health Carlos III (ISCIII), 28029 Madrid, Spain; 9Centro de Investigación Educativa y Bienestar Universitario, Departamento de Disciplinas Filosófico, Metodológicas e Instrumentales, División de Disciplinas Básicas, Centro Universitario de Ciencias de la Salud (CUCS), Universidad de Guadalajara (UdeG), Guadalajara 44340, Mexico; 10Centro de Investigación de Endocrinología y Nutrición Clínica, Universidad de Valladolid, 47005 Valladolid, Spain

**Keywords:** food selection, sociodemographic factors, updated food choice questionnaire, eating behavior

## Abstract

Several studies have explored food choice motives (FCMs), but their association with sociodemographic characteristics remains under-researched. This study aimed to examine the cross-sectional associations between sociodemographic factors and FCMs in a sample of Mexican adults. Sociodemographic data and eight FCMs’ importance (health and natural content, mood, convenience, sensory appeal, price, food identity, image, and environmental concern) measured with a 75-item Updated Food Choice Questionnaire were studied in 786 participants (70% women, mean age: 24.7 years). The adjusted estimates from multivariable linear regressions are reported. Higher relative importance of convenience, price, and image management (all, *p* < 0.05) were associated with men. Older participants (≥41 years) showed greater concern for environmental and wildlife awareness (EWA) (*p* = 0.04). Higher education was associated with higher appreciation of health and natural content and EWA (both, *p* < 0.05), and less of sensory appeal and mood (both, *p* < 0.05). Individuals in a relationship placed less importance on sensory appeal compared to singles (*p* = 0.008). Middle–low socioeconomic status was linked to higher importance of food identity (*p* = 0.039), while food insecurity was associated with higher concern for price and food identity (both, *p* < 0.05). Our findings highlight the influence of sociodemographic factors on FCMs that might act as barriers or drivers for adopting healthy diets.

## 1. Introduction

Food choice motives encompass a range of factors that influence individuals’ decisions regarding the selection and consumption of food [1]. These motives can include practical considerations such as convenience and cost, as well as more personal factors like health consciousness and sensory appeal [2]. Understanding these motives is essential for designing targeted interventions in public health nutrition, marketing strategies, and policy-making [3].

Sociodemographic factors such as sex, age, income, education level, marital status, among others, significantly shape individuals’ food choices [4]. However, these are often overlooked in research [5]. As an example, sex-related differences in this process have been documented, where women tend to consume more fruits and vegetables compared to men, as well as pay more attention to ethical considerations and engage with more frequency in dietary restrictions [6]. Similarly, age-related distinctions in food choice motives have also been identified. In this sense, older adults often prioritize health-related motives, while younger individuals may emphasize convenience and taste [5].

Since food choices partly determine dietary habits, a primary determinant of health status [7], it is crucial to explore how these motives are influenced by personal characteristics. This understanding is necessary for effectively monitoring contemporary food consumption and directing health actions for specific segments of the population [8]. While differences between sociodemographic groups and diet have been extensively studied, research examining sociodemographic factors and food choice motives remains scarce [5,8,9,10,11,12]. Existing studies indicate that, compared to men, women attach greater importance to health-related aspects and weight control [6,13,14,15]. Older individuals, in contrast to younger ones, place more value on motives related to long-term benefits, such as health, natural content, and environmental aspects [6,13,15]. Moreover, a lower socioeconomic status seems to be related to prioritizing price over health [13,14,16,17].

However, much of this research has been conducted in developed countries, with only a few studies from Brazil [18] and Uruguay [19] addressing the link between food choice motives and sociodemographic factors in Latin America. Both of them have primarily focused on how food choice motives vary according to socioeconomic status. In Mexico, a country where diet-related diseases are highly prevalent [20], the situation is even more complex. Limited research has explored food choices within this vulnerable population, often focusing on specific aspects such as identifying consumer segments [21], organic food consumption [22], and traditional food preferences [23]. Nonetheless, to our knowledge, data on sociodemographic differences in food choice motives in this country are non-existent.

Furthermore, most of studies regarding food choices have analyzed the absolute importance given to the specific motives [24,25,26,27,28,29]. A limitation of this approach is that it considers a broad range of potentially relevant motives, without addressing the need for individuals to prioritize them [30]. Consequently, it has been proposed that, in addition to evaluating the absolute importance, the relative importance (obtained by dividing the score given to each specific motive by the average of all the food choice motives) should also be taken into account [8]. This could better reflect the complex structure of motives involved in food choices. Therefore, the present study aimed to investigate the association between sociodemographic characteristics and the relative importance towards diverse food choice motives in Mexican adults.

## 2. Materials and Methods

### 2.1. Study Design, Ethical Approval, and Participants

This cross-sectional study included Mexican adults recruited by convenience sampling through two different research projects. Both projects were conducted according to the Declaration of Helsinki and were approved by the Research, Research Ethics and Biosafety Committees from the University Center for Health Sciences (or CUCS for its initials in Spanish—*Centro Universitario de Ciencias de la Salud*) from the University of Guadalajara (*Universidad de Guadalajara*), in Mexico. Informed consent was obtained from all subjects.

The first project explores the association between food insecurity and cardiometabolic health among university students and workers in the University of Guadalajara (approval advice code CI-02323, obtained on 18 February 2022). Students and workers 18 years of age or older from CUCS were invited to participate face-to-face at the university. We excluded pregnant or lactating women, and participants taking specific drugs such as corticosteroids, anabolic steroids, or antiretrovirals, that could alter cardiometabolic parameters. The second project describes the construct validation process of the Updated Food Choice Questionnaire (U-FCQ) (approval advice code CI-04023, obtained on 13 May 2023), the tool that measures food choice motives’ importance in this study. For that study, individuals from different states of Mexico were invited to participate through a poster that was distributed on social media. Participants were required to be 18 years of age or older, and to have access to a computer or electronic device to fill out online forms. Both samples are described with more detail elsewhere [31].

### 2.2. Measurements of Predictors

Participants from both projects completed an online self-administered questionnaire that collected sociodemographic characteristics (sex, age, educational level, and employment, socioeconomic, marital, food security, and housing status).

Age was categorized into 3 groups (18–29, 30–40, and 41 and over), according to Konttinen et al. [8], while the marital status of participants was defined as single (including divorced or widowed) or in a relationship (married or common law). The housing status was defined as house owner, tenant, or borrower. For the categorization of educational level, participants indicated their last academic degree obtained, and were classified as basic level (elementary, middle school, high school) or higher level (bachelor’s degree, graduate degree, or medical specialty).

To determine the socioeconomic status, we used the AMAI’s tool (for its acronym in Spanish: *Asociación Mexicana de Agencias de Inteligencia de Mercado y Opinión*), validated for the Mexican population [32]. This tool considers the educational level of the head of household, and the number of rooms in the household, complete bathrooms, people ≥ 14 years of age living in the household, cars, as well as access to the internet. According to the obtained score, 7 socioeconomic levels are defined, using letters: A/B (highest), C+, C, C−, D+, D, and E (lowest). For the current analysis, these levels were then grouped into two categories: high (A/B and C+) and middle or low socioeconomic status (all the other categories).

To assess the participants’ food security status, they were asked to answer the validated Latin American and Caribbean Scale for Food Security [33] (ELCSA, for its acronym in Spanish). This tool consists of 8 items or up to 15 items (if there are individuals under 18 years of age in the household). Through dichotomous questions (yes or no), it assesses if the participant, whether due to lack of money or other resources, has experienced worries about running out of food, not eating or eating less, not following a healthy diet, eating a diet with poor variety, etc. According to the number of obtained affirmations, a score is calculated and the participants are classified into one of the following categories, with different cut-off points in households without or with children under 18 years of age: food security (0 points), mild food insecurity (1–3 or 1–5 points, respectively), moderate food insecurity (4–6 or 6–10 points, respectively), or severe food insecurity (7–8 or 11–15 points, respectively).

### 2.3. Measurement of Outcomes

Finally, food choice motives were evaluated through the U-FCQ, a questionnaire adapted and validated for its use in the Mexican population [31]. Briefly, this tool, which was also completed online, consists of 75 items that assess the importance given to different food choice motives, using a 4-choice Likert-type scale (1 = not at all important to 4 = very important).

These motives are grouped into 8 dimensions or major motives: (1) health and natural content (27 items), which refers to the importance given to the effect of food and its components on the organism; (2) environmental and wildlife awareness (12 items), addressing the importance given to the effect of food production and transportation on the environment, and ethical treatment of animals; (3) convenience (6 items), covering aspects related to the time and ease of food preparation and consumption; (4) price (5 items), which includes economic aspects of food; (5) sensory appeal (6 items), regarding hedonic elements of eating, such as taste, smell, texture, and appearance of food; (6) mood (5 items), including aspects related to the effects of food selection on the emotional state; (7) image management (8 items), related to the importance given to the image projected to other people in relation to the food chosen, and the affinity for known foods; and (8) food identity (6 items), related to the use of food as a means for social interaction and the importance of food reflecting the individual’s culture.

In its original version, this tool allows the calculation of the absolute importance’s mean of each dimension, ranging from 1 to 4, where a higher score represents greater importance [13]. The mean is calculated from the mean of all the items considered in a specific dimension.

When assessing food choice motives, it is crucial to consider the factors that individuals consider as relevant [3]. Often, these motives are in conflict—for instance, prioritizing price over health—requiring individuals to make trade-offs [34]. Therefore, it has been suggested that, beyond evaluating absolute importance, examining relative importance of food choice motives provides additional insight. This approach divides the score of a specific motive by the mean score of all motives, offering a clearer understanding of how priorities are balanced [14]. In our study, to assess the relative importance of the U-FCQ dimensions, we employed the methodology proposed by Konttinen et al. [8]. This method consists in dividing, for each participant, the absolute mean (1 to 4) of each dimension by the mean of the 75 items of the tool. Dimensions with scores > 1 indicate that they are more important than all the assessed food choice motive items on average [8].

### 2.4. Statistical Analysis

Descriptive statistics were used to characterize the population. Age is presented as mean (standard deviation), while the categorical variables are presented as frequency (percentage). A *t*-test for independent samples and one-way ANOVA with Bonferroni’s post-hoc correction were used to compare quantitative variables between 2 or ≥3 groups, as appropriate, whereas qualitative variables were compared using the Chi-squared test.

Multivariable linear regression models were performed to evaluate the independent contributions of selected sociodemographic factors (predictor variables), accounting for their simultaneous inclusion in the model, for each food choice motive’ relative importance (outcome variables) analyzed separately. Two types of coding for age were employed: when age was a control variable, continuous coding was used (years), and when age was a predictor, categorical coding was used (age groups) to enable detection of nonlinear associations [8]. The following sociodemographic characteristics were selected based on the previous literature: sex (female (ref.) male), age (18–29 (ref.), 30–40, 18–29, ≥41 years), educational level (basic (ref.), higher), marital status (single/divorced/widowed (ref.), in a relationship), employment status (unemployed (ref.), employed), socioeconomic status (middle or low, high (ref.)), housing status (tenant, owner (ref.), borrower), and food security status (food security (ref.), mild food insecurity, moderate food insecurity, severe food insecurity). Multicollinearity was assessed by inspecting the tolerance (1/VIF) values and variance inflation factors (VIFs) for this multivariable model. All tolerance values were above 0.7 and all VIFs were below 2.0, which suggests there were no concerns over multicollinearity [35]. The Ramsey regression error specification (RESET) test was used to detect omitted variables and incorrect form in the linear regression model (null hypothesis > 0.05) [36]. Estimates were presented as coefficient (β) and 95% confidence interval (CI).

All statistical analyses were performed using STATA^®^ software, version 15.0 (Stata Corp., College Station, TX, USA). A *p*-value < 0.05 was considered statistically significant.

## 3. Results

### 3.1. Participants’ Characteristics

For the current analysis, all participants who had data regarding food choice motives were included. The total study sample therefore comprised 786 subjects (512 from the first project and 274 from the second one). The majority of the respondents were women (70.0%), and the most frequent age group was 18–29 years, with a mean age of 24.7 (8.5) years. Most of the participants were single (81.4%), had a basic educational level (68.1%), and were owners of their households (59.7%). Additionally, 51.3% of the participants experienced some degree of food insecurity. Compared to women, a higher proportion of men were employed (56.8 vs. 43.9, *p* = 0.002) and had a higher SES (48.3% vs. 38.6%, *p* = 0.012). Complete information on the characteristics of the respondents is presented in Table 1.


### 3.2. Food Choice Dimensions’ Relative Importance

For the entire sample, the highest food motives’ relative importance was placed on sensory appeal, convenience, mood, health and natural content, and price, whereas the lowest was for food identity, environmental and wildlife awareness, and image management (Table 2).


### 3.3. Associations Between Sociodemographic Characteristics and Relative Food Choice Motives’ Importance

Table 3 and Table 4 present the independent associations between sociodemographic factors (predictor variables) and each relative food choice motive (outcome variables) analyzed separately. Specifically, Table 3 includes the models for health and natural content, environmental and wildlife awareness, convenience, and price dimensions, while Table 4 covers sensory appeal, mood, image management, and food identity.

In the multivariable linear regression models, men were associated with placing lower importance on health and nutritional content (*p* = 0.023) and environmental and wildlife awareness (*p* < 0.001) compared to women. Conversely, men attached higher importance to convenience (*p* = 0.008), price (*p* < 0.001), and image management (*p* = 0.004). On the other hand, participants aged ≥41 showed greater concern for environmental and wildlife awareness, while those aged 30–40 placed higher importance on image management than the younger group (18–29 years) (Table 3).

In terms of education, a higher level was associated with a greater importance of health and natural content (*p* = 0.011) and awareness of the environment and wildlife (*p* = 0.044) (Table 3). On the contrary, it was associated with lower importance of sensory appeal (*p* = 0.035) and mood (*p* = 0.030) compared to basic educational level (Table 4).

For marital status, those with a partner gave lower importance to sensory appeal (*p* = 0.008) and higher to food identity (*p* = 0.017) than singles. Similarly, employed participants attached lower importance to sensory appeal (*p* = 0.002) and higher importance to food identity (*p* = 0.001) (Table 4).

Lower–middle socioeconomic status was associated with a higher importance of food identity (*p* = 0.039) compared to the high socioeconomic status group. Renters prioritized convenience (*p* = 0.031) but gave less importance to image management (*p* = 0.031) than owners. Lastly, mild food insecurity was associated with greater importance of price (*p* = 0.013), while the moderate category was linked to greater importance of price (*p* = 0.08) and food identity (*p* = 0.009), when compared to food security groups (Table 4).


## 4. Discussion

This paper provides a comprehensive analysis of the differences in food choice motives’ importance across various sociodemographic characteristics in Mexican adults. Sex, age, educational level, and marital status emerged as the main factors showing association with certain food choice motives’ dimensions.

Regarding sex differences, men placed less importance on environmental and wildlife awareness, and more value on price, convenience, and image management in comparison to women. This pattern aligns with existing studies carried out in adults from Switzerland [9] and Saudi Arabia [11] that suggest sex differences in food behavior. The previous literature has shown that women are generally more environmentally conscious and more likely to engage in pro-environmental behaviors [38,39]. This may be explained by several reasons. On the one hand, environmental care is generally seen as “feminine,” which may make women more likely to identify with these values [40]. On the other hand, women often assume responsibility for food shopping and food preparation in the household [41]. This role may increase their concern about how food is produced, as well as the impact of its consumption on the environment. In addition, women are more likely to be exposed to media and literature that highlights environmental issues and animal rights [40]. This increased exposure may shape their attitudes and influence their prioritization in this area. Furthermore, gender norms associating masculinity with certain types of food—such as red meat and fast food—may further reinforce these patterns [40,42].

Regarding men’s higher appreciation for price and convenience, this could be reflective of traditional gender roles and societal expectations as providers [43,44]. Despite the social changes experienced in Mexico in recent years, as a result of the increasing integration of women into the workforce, men have been described as the ones who make most of the important decisions about family expenses [45]. Therefore, this can explain the prioritization of cost and convenience aspects due to work-related time constraints [46]. The fact that men gave greater importance to image management may be due to a variety of circumstances. In many societies, including Mexico, men are encouraged to project strength, success, and independence [47], which can influence their food choices. For example, men may select food that aligns with a socially constructed image of masculinity, such as protein-heavy meals or food marketed as enhancing physical performance or fitness [48]. Additionally, food marketing often targets men with messaging that connects food choices to attributes such as power, confidence, or attractiveness [49]. This type of signaling reinforces the notion that food can be a tool for projecting a desirable image [48]. Finally, there is still the stigma attached to foods perceived as “feminine” (e.g., salads or low-calorie meals), which may deter men from choosing these options [50]. Instead, they may opt for foods that project a more socially acceptable image, even if those foods do not align with other priorities like health or sustainability.

Participants with greater age (≥41 years) showed higher importance on environmental and wildlife awareness in comparison with the 18–29 category. Our results differ from the previous literature which reports that younger people often are more environmentally educated and concerned than older people [38,51]. This observed trend can be interpreted trough several lenses. First, our findings may reflect generational differences in values and life experiences. For example, individuals in the older age group may have been exposed to environmental issues for a longer period, and their awareness of the long-term consequences of environmental degradation could be more acute [38]. This heightened awareness could be due to their greater exposure to environmental movements, conservation efforts, and public discourse on sustainability over the decades, which may have shaped their attitudes towards environmental responsibility. Additionally, older adults may have more financial stability, which allows them to make food choices that align with their environmental values, even if these options are more expensive or less convenient.

In contrast, younger age groups, particularly those in the 18–29 category, may prioritize factors such as convenience or price due to different life circumstances, such as lower income or busier lifestyles [52]. On the other hand, the literature shows that, as individuals age, they tend to make food choices oriented by long-term effects, such as their overall health and sustainability [8,15], whereas younger individuals are more prone to make food choices motivated by immediate effects like sensory appeal, mood, price, and convenience [8]. This pattern of food choice motives can be explained by social differences between adults between 18 and 29 years of age and those who are older. For example, in Mexico, from age 30 onwards, it is more common to be married, have children, exhibit risk factors for chronic diseases, or have family members with these conditions, all of which can lead to greater health consciousness [6,45,53].

Higher educational level was linked to a greater value of health and natural content and environmental and wildlife awareness, as well as with lower appreciation for sensory appeal and mood. The emphasis on health and natural content among highly educated individuals can be attributed to increased awareness and knowledge about the benefits of healthy eating and the potential risks associated with processed foods [54]. Education often equips individuals with the ability to critically evaluate nutritional information and understand the long-term health implications of their dietary choices [55]. This is consistent with previous research indicating that higher educational attainment is associated with healthier dietary habits [56,57], and with the willingness to pay for healthier food options [58]. Thus, the lower appreciation for sensory appeal and mood among the more educated may reflect that these individuals are more willing to invest in higher-quality, health-oriented products, even if they come at a higher cost or may not always meet their sensory expectations [58].

In terms of marital status, its association with eating behavior and nutritional status has been previously described [46]. For example, married individuals have higher rates of vegetable and fruit consumption, as well as a lower risk of diet-related diseases [10,12,53]. However, our study did not find significant results between marital status and the importance given to motives related to health and natural content, which reflects the degree of concern for the consumption of foods that promote a beneficial effect on the organism. This could be partly due to the young age of most of the participants (on average 24 years; 80% are in the age category 18–29 years), and, as previously mentioned, young people tend to prioritize other aspects when making food choices [5]. On the other hand, we found that participants in a relationship (married or cohabiting) gave less importance to sensory attractiveness than single people. This is in line with the findings in the study by Eng and colleagues who reported that widowed or divorced individuals decreased their consumption of vegetables and increased their consumption of fried foods away from home, which may be perceived as more palatable [59]. One possible explanation for this finding is that individuals in relationships might prioritize other factors, such as compatibility, shared goals, and emotional support, over sensory attributes [44]. Moreover, the act of eating together and engaging in conversation can be more important than the specific taste, smell, or textures of foods [60].

Specifically, we observed that individuals from lower–middle socioeconomic status placed significantly more emphasis on food identity compared to those in the higher socioeconomic group. This outcome may suggest that for individuals with more limited economic resources, food identity serves as a crucial element of self-expression and cultural connection, possibly due to less accessibility of diverse culinary options or greater reliance on familiar and culturally resonant foods [23]. The alignment of food choices with personal or group identity might thus serve as a compensatory mechanism, providing a sense of belonging and self-worth [61,62]. In addition, the greater importance of food identity in this group may be due to the reliability on foods that are more economically sensitive, such as those included in the basic food basket, which are considered traditional [52].

Furthermore, this study highlights the critical role of food insecurity in determining food choice motives. Individuals experiencing mild food insecurity were more likely to prioritize price, which is consistent with the existing literature indicating that economic constraints strongly influence food choice in this sociodemographic group [63], particularly in Mexican households [64]. Interestingly, moderate food insecurity was associated not only with a greater emphasis on price, but also on food identity. This dual focus may indicate that as food insecurity becomes more severe, individuals may seek to maintain a connection to their food identity even as financial constraints become more pronounced [23]. The prioritization of food identity in this context could represent an attempt to preserve cultural and personal food-related practices despite the challenges posed by limited resources, as well as the selection of food perceived as more economically accessible [23,52].

The present study has several strengths. To the best of our knowledge, this is the earliest study to fully focus on the association between food choice motives and sociodemographic characteristics among Mexican participants. In our country, a study by Serrano-Cruz and colleagues addressed certain aspects of food choices [23]. However, they only used five out of the thirty-six items of the original FCQ [13]. Additionally, its aim was to determine the association between sociodemographic factors and the specific consumption of traditional foods. In contrast, our research considered a broader spectrum of food choice motives. On the other hand, to assess food choices, we used a questionnaire specifically adapted and validated for the Mexican population by our research group [31]. This tool, in contrast to other versions of the FCQ, includes more items that allow for a more in-depth evaluation of this process. Finally, our study analyzed the relative importance of food choice motives. Using this methodology allowed us to compare the importance given to the U-FCQ dimensions with each other. For instance, while some individuals may find several motives relevant, relative importance reveals which motive carries more weight in their decision-making. Moreover, absolute importance often results in multiple motives being rated as “very important”, making it difficult to discern clear priorities [65]. Relative importance normalizes these scores, enabling more meaningful and realistic understanding of food choice dynamics [8,14].

Despite the above, some limitations should be considered. For instance, our sample included a higher proportion of women than men. This can be attributed to a number of factors. Women generally exhibit greater interest in health-related issues, potentially due to traditional roles as caregivers and responsibilities as household managers [41]. Additionally, it has been suggested that women are more likely to participate in surveys and research due to higher levels of altruism and greater interest in contributing to science [66]. These factors should be considered when interpreting the study findings and planning future research to ensure a more balanced representation of sexes.

On the other hand, most of our participants (80%) were in the category of 18 to 29 years of age, with an underrepresentation of those over 40 years of age. This may have been due to the fact that recruitment was conducted within a university context and trough social media, which tend to have greater reach and engagement among younger adults. It is possible that this demographic imbalance may have influenced the generalizability of our findings. For example, food choice priorities may vary across life stages and could be shaped by factors such as health concerns, economic stability, and cultural norms, all of which are more pronounced in older age groups [5,67]. Thus, we recommend that future studies ensure a more balanced age distribution to explore food choice behaviors more comprehensively among all age groups.

Moreover, it is important to emphasize that more than half of the participants come from a project carried out among university students in the health area [68]. In this sense, it is possible that due to the nature of their undergraduate studies, the importance of food choice motives has shifted towards health-related aspects and overshadowed others such as the environment and wildlife awareness. Recognizing the latter is important to accurately interpret these results.

Other statistical methods, such as principal component analysis or cluster analysis, could provide deeper insight into the structure of food choice motives and could help disentangle the underlying relationships between these factors. However, the current study was focused on examining specific associations. Subsequent studies could follow a more exploratory approach. Furthermore, several other factors undoubtedly influence the food choice process, some of which, due to the nature of this tool, we were unable to explore. These factors include food availability, exposure to food advertising, peer influence, and biological determinants. Future research may benefit from employing complementary methodologies to better assess these elements and explore their potential influence and interactions on food choice motives.

The results obtained from this study are intended to serve as a resource for decision-makers. Recognizing the sociodemographic differences in food choice motives provides an opportunity to design interventions that address specific needs and preferences while promoting healthier and more sustainable diets. For instance, interventions can capitalize on the observed sex differences in food choice motives to create culturally relevant health messages. Men, in particular, may respond well to initiatives that help them make healthier food choices that align with price, convenience, and positive image management. Strategies such as subsidies for fruits, vegetables, and nutritious foods, along with the promotion of simple, cost-effect recipes, could support healthier eating habits. For younger populations, interventions should connect healthy and environmentally friendly food choices to current trends, such as plant-based diets and fitness culture, while showcasing these choices as modern, innovative, and aligned with popular culture. Rather than focusing on long-term health or environmental outcomes (which might feel abstract to them), strategies could emphasize immediate benefits, such as improved energy, better skin, enhanced athletic performance, or even cost savings. In particular, sociodemographic groups experiencing food insecurity could benefit from recommendations promoting the consumption of affordable, healthy traditional foods, which consider the economic constraints faced by individuals.

## 5. Conclusions

Our findings confirm that, in a Mexican context, sex, age, educational level, marital status, employment activity, socioeconomic level, and household food insecurity are significant determinants of food choices. These factors can either act as barriers or facilitators for adopting healthy dietary habits. This study underscores the importance of considering sociodemographic characteristics when designing nutrition interventions. By recognizing and addressing the needs and priorities specific to each group, policymakers and health promoters can enhance the effectiveness of strategies aimed at fostering healthier, more sustainable diets for all segments of the population.

## Figures and Tables

**Table 1 foods-14-00158-t001:** Descriptive characteristics of study participants.

Parameters	Total (*n* = 786)	Men (*n* = 236)	Women (*n* = 550)	*p*-Value ^1^
Age, mean (SD)	24.7 (8.5)	24.5 (7.4)	24.8 (8.9)	0.609
Age group, *n* (%)				
18–29 years old	629 (80.0)	185 (78.4)	444 (80.7)	**0.034**
30–40 years old	109 (13.9)	42 (17.8)	67 (12.2)	
≥41 years old	48 (6.1)	9 (3.8)	39 (7.1)	
Marital status, *n* (%)				
Single, divorced, or widowed	640 (81.4)	195 (82.6)	445 (80.9)	0.570
In a relationship ^2^	146 (18.6)	41 (17.4)	105 (19.1)	
Educational level, *n* (%)				
Basic level ^3^	535 (68.1)	152 (64.4)	383 (69.8)	0.140
Higher level ^4^	250 (31.9)	84 (35.6)	166 (30.2)	
Employment status, *n* (%)				
Unemployed	410 (52.3)	102 (43.2)	308 (56.1)	**0.002**
Employed	375 (47.7)	134 (56.8)	241 (43.9)	
SES ^5^, *n* (%)				
Middle or low	459 (58.5)	122 (51.7)	337 (61.4)	**0.012**
High	326 (41.5)	114 (48.3)	212 (38.6)	
Housing status, *n* (%)				
Owner	469 (59.7)	143 (60.6)	326 (59.3)	0.891
Tenant	238 (30.3)	71 (30.1)	167 (30.4)	
Borrower or other	79 (10.0)	22 (9.3)	57 (10.4)	
Food security status ^6^, *n* (%)				
Food security	381 (48.7)	115 (48.7)	266 (48.4)	0.755
Mild food insecurity	228 (29.0)	65 (27.5)	163 (29.6)	
Moderate food insecurity	109 (13.9)	37 (15.7)	72 (13.1)	
Severe food insecurity	68 (8.7)	19 (8.1)	49 (8.9)	
Geographic residential area ^7^, *n* (%)				
North	101 (12.8)	39 (16.5)	62 (11.3)	0.164
West	674 (85.8)	191 (81.0)	483 (87.8)	
Center and southeast	11 (1.4)	6 (2.5)	5 (0.9)	

Values are expressed as mean (SD) or frequency (%). Abbreviations—SD: standard deviation; SES: socioeconomic status. ^1^ *p*-values were calculated with the Chi-squared test (for qualitative variables) or Student’s *t*-test (for age). The significance of the numbers in bold is *p*-value < 0.05. ^2^ Married or common law. ^3^ Elementary, middle school, high school. ^4^ Bachelor’s degree, graduate degree, or medical specialty. ^5^ Assessed through the AMAI’s (*Asociación Mexicana de Agencias de Inteligencia de Mercado y Opinión*) tool [32]. The original 7 categories (A/B, C+, C, C−, D+, D, and E) were dichotomized into high (A/B and C+) and middle–low socioeconomic status (all the other categories). ^6^ Determined through the Latin American and Caribbean Scale for Food Security [33] (ELCSA for its acronym in Spanish). ^7^ Categories derived from the geographic distribution of Mexico: northern region, western region, central region, and southeastern region (the latter two were combined for statistical purposes) [37].

**Table 2 foods-14-00158-t002:** Mean values (SD) of relative food choice motives’ importance in the whole sample and by sociodemographic groups.

Variable	*n* (%)	Sensory Appeal	Mood	Convenience	HNC	Price	Food Identity	EWA	Im
		Mean (SD)							
Total sample	786 (100.0)	1.24 (0.21)	1.09 (0.18)	1.10 (0.19)	1.08 (0.10)	1.06 (0.23)	0.90 (0.16)	0.85 (0.15)	0.63 (0.15)
Sex									
Male	236 (30)	1.25 (0.23)	1.07 (0.20)	1.12 (0.21)	1.08 (0.10)	1.10 (0.26)	0.90 (0.17)	0.82 (0.16)	0.66 (0.15)
Female	550 (70)	1.24 (0.20)	1.10 (0.17)	1.08 (0.18)	1.09 (0.09)	1.04 (0.21)	0.90 (0.16)	0.86 (0.14)	0.62 (0.16)
*p*-value ^1^		0.577	0.116	**0.012**	0.357	**<0.001**	0.462	**<0.001**	**0.008**
Age group									
18–29 years old	629 (80.0)	1.25 (0.21) *	1.10 (0.19) *	1.10 (0.20)	1.08 (0.10)	1.07 (0.24) *	0.90 (0.16)	0.84 (0.15) *	0.64 (0.15)
30–40 years old	109 (13.9)	1.20 (0.19) *	1.07 (0.14)	1.10 (0.15)	1.10 (0.09)	1.02 (0.19)	0.90 (0.16)	0.87 (0.13)	0.61 (0.16)
≥41 years old	48 (6.1)	1.16 (0.22) *	1.02 (0.13) *	1.07 (0.13)	1.10 (0.08)	0.96 (0.17) *	0.94 (0.13)	0.93 (0.12) *	0.60 (0.18)
*p*-value ^1^		**0.001**	**0.016**	0.532	0.051	**0.001**	0.201	**<0.001**	0.055
Marital status									
Single	640 (81.4)	1.26 (0.21)	1.10 (0.19)	1.10 (0.20)	1.08 (0.10)	1.07 (0.23)	0.89 (0.16)	0.84 (0.15)	0.63 (0.15)
In a relationship ^2^	146 (18.6)	1.17 (0.18)	1.04 (0.15)	1.08 (0.15)	1.09 (0.08)	1.02 (0.21)	0.93 (0.13)	0.89 (0.14)	0.63 (0.17)
*p*-value ^1^		**<0.001**	**<0.001**	0.176	0.131	**0.020**	**0.008**	**<0.001**	0.890
Educational level									
Basic ^3^	536 (68.1)	1.26 (0.21)	1.11 (0.19)	1.11 (0.21)	1.07 (0.10)	1.08 (0.24)	0.90 (0.15)	0.83 (0.15)	0.64 (0.15)
Higher ^4^	250 (31.9)	1.20 (0.19)	1.05 (0.15)	1.08 (0.16)	1.10 (0.09)	1.01 (0.18)	0.90 (0.16)	0.87 (0.14)	0.63 (0.17)
*p*-value ^1^		**<0.001**	**<0.001**	**0.045**	**<0.001**	**<0.001**	0.710	**<0.001**	0.557
SES ^5^									
Middle or low	459 (58.5)	1.25 (0.21)	1.09 (0.19)	1.10 (0.20)	1.08 (0.09)	1.08 (0.24)	0.91 (0.15)	0.84 (0.15)	0.63 (0.15)
High	326 (41.5)	1.23 (0.21)	1.09 (0.18)	1.09 (0.18)	1.09 (0.10)	1.03 (0.18)	0.90 (0.16)	0.85 (0.15)	0.63 (0.16)
*p*-value ^1^		0.404	0.874	0.847	0.153	**0.002**	0.342	0.333	0.999
Food security status ^6^									
Food security	381 (48.5)	1.25 (0.21)	1.08 (0.19)	1.09 (0.18)	1.09 (0.09)	1.03 (0.21) *	0.90 (0.16)	0.84 (0.15)	0.63 (0.15)
Mild FI	228 (29.0)	1.23 (0.20)	1.09 (0.18)	1.09 (0.19)	1.08 (0.09)	1.07 (0.23)	0.93 (0.15) *	0.85 (0.14)	0.63 (0.15)
Moderate FI	109 (13.9)	1.21 (0.16)	1.12 (0.16)	1.13 (0.19)	1.08 (0.09)	1.10 (0.24) *	0.86 (0.17) *	0.84 (0.15)	0.64 (0.16)
Severe FI	68 (8.7)	1.28 (0.30)	1.09 (0.19)	1.11 (0.26)	1.07 (0.13)	1.08 (0.28)	0.88 (0.16)	0.84 (0.18)	0.65 (0.16)
*p*-value ^1^		0.102	0.459	0.239	0.368	**0.012**	**0.001**	0.825	0.756

Values are expressed as mean (SD). Abbreviations—HNC: health and natural content; EWA: environmental and wildlife awareness; Im: image management; SD: standard deviation; SES: socioeconomic status; FI: food insecurity. ^1^ *p*-values were calculated with Student’s *t*-test or ANOVA test, as appropriate. The significance of the numbers in bold is *p*-value < 0.05. Categories in which significant differences were observed after ANOVA with post-hoc Bonferroni correction are marked with * (*p* < 0.05). ^2^ Married or common law. ^3^ Elementary, middle school, high school. ^4^ Bachelor’s degree, graduate degree, or medical specialty. ^5^ Assessed through the AMAI’s (Asociación Mexicana de Agencias de Inteligencia de Mercado y Opinión) tool [32]. The original 7 categories (A/B, C+, C, C−, D+, D, and E) were dichotomized into high (A/B and C+) and middle–low socioeconomic status (all the other categories). ^6^ Determined through the Latin American and Caribbean Scale for Food Security [33] (ELCSA for its acronym in Spanish).

**Table 3 foods-14-00158-t003:** Multivariable linear regression models of associations between sociodemographic characteristics and relative food motives’ importance: health and natural content, environmental and wildlife awareness, convenience, and price (*n* = 786).

Sociodemographic Characteristics	Health and Natural Content			Environmental and Wildlife Awareness			Convenience			Price		
	β ^1^	(95% CI)	*p* ^2^	β	(95% CI)	*p* ^2^	β	(95% CI)	*p* ^2^	β	(95% CI)	*p* ^2^
Sex												
Male vs. female (ref.)	−0.01	−0.03, 0.01	**0.023**	−0.05	−0.07, −0.02	**<0.001**	0.04	0.01, 0.07	**0.008**	0.08	0.04, 0.11	**<0.001**
Age group												
30–40 vs. 18–29 y (ref.)	0.00	−0.02, 0.03	0.753	0.02	−0.03, 0.06	0.460	0.03	−0.02, 0.08	0.253	−0.03	−0.09, 0.03	0.391
≥41 vs. 18–29 y (ref.)	0.01	−0.03, 0.05	0.576	0.06	0.00, 0.11	**0.044**	0.01	−0.06, 0.08	0.832	−0.07	−0.15, 0.01	0.087
Educational level												
Superior ^3^ vs. Basic ^4^ (ref.)	0.02	0.01, 0.04	**0.011**	0.01	−0.01, 0.04	0.330	−0.04	−0.07, 0.00	0.065	−0.04	−0.09, 0.00	**0.045**
Marital status												
In a relationship vs. Single, divorced or widowed (ref.)	−0.01	−0.03, 0.02	0.514	0.02	−0.02, 0.05	0.348	−0.01	−0.06, 0.04	0.627	0.02	−0.03, 0.08	0.424
Employment status												
Employed vs. unemployed (ref.)	0.01	−0.00, 0.02	0.158	0.02	−0.00, 0.04	0.080	−0.01	−0.04, 0.02	0.488	−0.02	−0.05, 0.01	0.243
SES ^5^												
Middle or low vs. high (ref.)	−0.01	−0.01, 0.03	0.197	0.00	−0.02, 0.03	0.760	−0.01	−0.04, 0.02	0.624	0.03	−0.01, 0.07	0.062
Housing status												
Tenant vs. owner (ref.)	0.00	−0.02, 0.02	0.910	−0.01	−0.04, 0.01	0.324	0.03	0.00, 0.07	**0.031**	0.01	−0.03, 0.05	0.561
Borrower vs. owner (ref.)	−0.01	−0.03, 0.02	0.738	0.00	−0.03, 0.04	0.949	−0.01	−0.05, 0.04	0.792	0.02	−0.04, 0.07	0.497
Food security status ^6^												
Mild FI vs. food security (ref.)	−0.01	−0.03, 0.00	0.123	0.00	−0.02, 0.30	0.857	−0.00	−0.03, 0.03	0.947	0.05	0.01, 0.09	**0.013**
Moderate FI vs. food security (ref.)	−0.01	−0.03, 0.01	0.383	−0.00	−0.04, 0.03	0.791	0.04	−0.01, 0.08	0.085	0.07	0.02, 0.11	**0.008**
Severe FI vs. food security (ref.)	−0.02	−0.05, 0.01	0.138	0.00	−0.04, 0.04	0.983	0.02	−0.03, 0.07	0.483	0.05	−0.01, 0.11	0.094
Goodness-of-fit measures	R^2^: 0.026, Adjusted R^2^: 0.016, Root MSE: 0.100			R^2^: 0.044, Adjusted R^2^: 0.034, Root MSE: 0.152			R^2^: 0.018, Adjusted R^2^: 0.008, Root MSE: 0.196			R^2^: 0.062, Adjusted R^2^: 0.052, Root MSE: 0.227		

Linear regression models were used to calculate the β coefficient (β) and 95% confidence interval (95% CI). For all models, VIFs were below 2.0, tolerance values were above 0.7, and no omitted variables were found according to the Ramsey RESET test. Abbreviations—U-FCQ: Updated Food Choice Questionnaire; CI: confidence interval; SES: socioeconomic status; FI: food insecurity; Root MSE: root mean square error. ^1^ The models were run separately for each food choice motive outcome. The models were mutually adjusted for all sociodemographic characteristics displayed in these tables as the exposure. ^2^ The significance of the numbers in bold is *p*-value < 0.05. ^3^ Bachelor’s degree, graduate degree, or medical specialty. ^4^ Elementary, middle school, high school. ^5^ Assessed through the AMAI’s (Asociación Mexicana de Agencias de Inteligencia de Mercado y Opinión) tool [32]. The original 7 categories (A/B, C+, C, C−, D+, D, and E) were dichotomized into high (A/B and C+) and middle–low socioeconomic status (all the other categories). ^6^ Determined through the Latin American and Caribbean Scale for Food Security [33] (ELCSA for its acronym in Spanish).

**Table 4 foods-14-00158-t004:** Multivariable linear regression models of associations between sociodemographic characteristics and relative food motives: sensory appeal, mood, image management, and food identity (*n* = 786).

Sociodemographic Characteristics	Sensory Appeal			Mood			Image Management			Food Identity		
	β ^1^	(95% CI)	*p* ^2^	β	(95% CI)	*p* ^2^	β	(95% CI)	*p* ^2^	β	(95% CI)	*p* ^2^
Sex												
Male vs. female (ref.)	0.01	−0.02, 0.05	0.386	−0.02	−0.05, 0.00	0.138	0.04	0.01, 0.06	**0.004**	−0.01	−0.04, −0.01	0.421
Age group												
30–40 vs. 18–29 y (ref.)	0.01	−0.05, 0.67	0.710	0.03	−0.02, 0.08	0.258	−0.05	−0.09, −0.01	**0.011**	−0.03	−0.07, 0.02	0.206
≥41 vs. 18–29 y (ref.)	0.00	−0.75, 0.077	0.984	−0.02	−0.08, 0.05	0.626	−0.04	−0.09, 0.04	0.080	0.01	−0.06, 0.06	0.934
Educational level												
Superior ^3^ vs. Basic ^4^ (ref.)	−0.04	−0.08, −0.00	**0.035**	−0.04	−0.07, 0.00	**0.030**	0.01	−0.02, 0.04	0.502	−0.01	−0.04, 0.02	0.405
Marital status												
In a relationship vs. Single, divorced or widowed (ref.)	−0.07	−0.11, −0.02	**0.008**	−0.04	−0.08, 0.00	0.079	0.04	−0.00, 0.07	0.054	0.05	−0.00, 0.09	**0.017**
Employment status												
Employed vs. unemployed (ref.)	−0.05	−0.08, −0.02	**0.002**	−0.02	−0.05, 0.00	0.176	−0.02	−0.04, 0.00	0.106	0.04	0.02, 0.07	**0.001**
SES ^5^												
Middle or low vs. high (ref.)	0.02	−0.05, 0.01	0.219	−0.03	−0.06, 0.00	0.058	−0.00	−0.03, 0.20	0.773	0.03	0.00, 0.05	**0.039**
Housing status												
Tenant vs. owner (ref.)	0.01	−0.03, 0.04	0.674	0.03	−0.00, 0.06	0.052	−0.03	−0.05, −0.00	**0.031**	−0.02	−0.04, 0.00	0.166
Borrower vs. owner (ref.)	0.01	−0.04, 0.06	0.700	0.02	−0.03, 0.06	0.507	−0.03	−0.07, 0.00	0.137	0.02	−0.02, 0.06	0.269
Food security status ^6^												
Mild FI vs. food security (ref.)	−0.01	−0.04, 0.03	0.608	0.01	−0.02, 0.04	0.534	−0.00	−0.03, 0.02	0.900	0.02	−0.00, 0.05	0.170
Moderate FI vs. food security (ref.)	−0.04	−0.08, 0.00	0.089	0.03	−0.00, 0.07	0.094	0.01	−0.02, 0.05	0.458	0.04	−0.08, −0.01	**0.009**
Severe FI vs. food security (ref.)	0.03	−0.03, 0.08	0.371	−0.00	−0.05, 0.04	0.881	0.02	−0.02, 0.06	0.252	−0.03	−0.07, 0.02	0.227
Goodness-of-fit measures	R^2^: 0.046, Adjusted R^2^: 0.036, Root MSE: 0.211			R^2^: 0.033, Adjusted R^2^: 0.023, Root MSE: 0.185			R^2^: 0.035, Adjusted R^2^: 0.025, Root MSE: 0.155			R^2^: 0.029, Adjusted R^2^: 0.019, Root MSE: 0.162		

Linear regression models were used to calculate the β coefficient (β) and 95% confidence interval (95% CI). For all models, VIFs were below 2.0, tolerance values were above 0.7, and no omitted variables were found according to the Ramsey RESET test. Abbreviations—U-FCQ: Updated Food Choice Questionnaire; CI: confidence interval; SES: socioeconomic status; FI: food insecurity; Root MSE: root mean square error. ^1^ The models were run separately for each food choice motive outcome. The models were mutually adjusted for all sociodemographic characteristics displayed in these tables as the exposure. ^2^ The significance of the numbers in bold is *p*-value < 0.05. ^3^ Bachelor’s degree, graduate degree, or medical specialty. ^4^ Elementary, middle school, high school. ^5^ Assessed through the AMAI’s (Asociación Mexicana de Agencias de Inteligencia de Mercado y Opinión) tool [32]. The original 7 categories (A/B, C+, C, C−, D+, D, and E) were dichotomized into high (A/B and C+) and middle–low socioeconomic status (all the other categories). ^6^ Determined through the Latin American and Caribbean Scale for Food Security [33] (ELCSA for its acronym in Spanish).

## Data Availability

The original contributions presented in the study are included in the article; further inquiries can be directed to the corresponding authors.

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
