# Peer review of "Associations of Sociodemographic Characteristics with Food Choice Motives’ Importance Among Mexican Adults: A Cross-Sectional Analysis"

_foods, 2025, doi:10.3390/foods14020158_

Round 1
Reviewer 1 Report
Comments and Suggestions for Authors
1. Abstract, Line 31, it is suggested to change it to "This study aimed to examine the cross-sectional associations between sociodemographic factors and food choice. motives in a sample of Mexican adults ", making it more natural to connect with the following content.
2. Introduction, it is recommended to streamline the background, focusing on key points related to the study and reducing unrelated descriptions.
3. Methods, Basic information about the sample (e.g., gender ratio, age range, education level) is provided, but the description of the sample is too brief. It is suggested to provide a more detailed account of the sample selection criteria, including the geographic distribution and selection standards, and discuss the representativeness of the sample.
4. The discussion of gender differences could be more in-depth. Please expand on the potential social and cultural factors that might explain gender-based variations in food choices.
5. Discussion, please add a section discussing how the results could inform specific health intervention strategies, particularly for vulnerable populations like those in Mexico.
Author Response
Subject: Reply to comments of reviewers regarding the manuscript “Associations of sociodemographic characteristics with food choice motives’ importance among Mexican adults: a cross-sectional analysis”
Date: December 26th, 2024
To the reviewers of Foods
Manuscript Foods-3373288
Respected reviewers,
On behalf of the research team, we thank you for your valuable comments on our work. This review process is crucial to improve our manuscript.
In the next lines, you fill find a point-by-point response at each one of your questions/observations. It is important to notice that all references to specific lines modifications relate to the .pdf file, to avoid confusions regarding different Office layouts or configurations. We hope that with these improvements our manuscript is now enough suitable for publication.
Kind regards,
Corresponding authors.
Reviewer #1
- Abstract, Line 31, it is suggested to change it to "This study aimed to examine the cross-sectional associations between sociodemographic factors and food choice. motives in a sample of Mexican adults ", making it more natural to connect with the following content.
Thank you for your suggestion. We have modified this in lines 31-32.
- Introduction, it is recommended to streamline the background, focusing on key points related to the study and reducing unrelated descriptions.
Thank you for your recommendation. We have modified the introduction to emphasize the relevant background to this research (lines: 64-92), as well as to make clearer the identified knowledge gap.
- Methods, Basic information about the sample (e.g., gender ratio, age range, education level) is provided, but the description of the sample is too brief. It is suggested to provide a more detailed account of the sample selection criteria, including the geographic distribution and selection standards, and discuss the representativeness of the sample.
Thank you for your observation. A more detailed account of the sample selection criteria and participants recruitment process were included in lines 106-116. In addition, data regarding the geographic distribution of the participants was included in Table 1.
- The discussion of gender differences could be more in-depth. Please expand on the potential social and cultural factors that might explain gender-based variations in food choices.
Thank you for your insightful observation. We have expanded the discussion related to the factors that could explain the differences between the importance given to different food choice motives according to sex in lines 323-334, 340-352, and 449-455.
- Discussion, please add a section discussing how the results could inform specific health intervention strategies, particularly for vulnerable populations like those in Mexico.
Thank you for your recommendation. We have included a section on how our results could be used to develop specific strategies to promote healthy diets in certain sociodemographic groups in lines 482-499.
Reviewer 2 Report
Comments and Suggestions for Authors
This paper sets out to explore the sociodemographic factors associated with food choice in Mexican adults. Using a self-completed on-line questionnaire the authors identify variation within the sample on the rationale for their choices.
The authors conclude that a number of factors are important in choice, for example sex, age and educational level and explain the possible implications for policy development towards helping achieve a more healthy diet.
The paper is well written and argued with pertinent analyses of the findings. The one small issue that the authors may like to consider commenting on centres on the small number of respondents aged 41 or above. That aside the authors are to be congratulated on their important work and the manner of its presentation.
Author Response
Subject: Reply to comments of reviewers regarding the manuscript “Associations of sociodemographic characteristics with food choice motives’ importance among Mexican adults: a cross-sectional analysis”
Date: December 26th, 2024
To the reviewers of Foods
Manuscript Foods-3373288
Respected reviewers,
On behalf of the research team, we thank you for your valuable comments on our work. This review process is crucial to improve our manuscript.
In the next lines, you fill find a point-by-point response at each one of your questions/observations. It is important to notice that all references to specific lines modifications relate to the .pdf file, to avoid confusions regarding different Office layouts or configurations. We hope that with these improvements our manuscript is now enough suitable for publication.
Kind regards,
Corresponding authors.
Reviewer #2
This paper sets out to explore the sociodemographic factors associated with food choice in Mexican adults. Using a self-completed on-line questionnaire the authors identify variation within the sample on the rationale for their choices.
The authors conclude that a number of factors are important in choice, for example sex, age and educational level and explain the possible implications for policy development towards helping achieve a more healthy diet.
The paper is well written and argued with pertinent analyses of the findings. The one small issue that the authors may like to consider commenting on centres on the small number of respondents aged 41 or above. That aside the authors are to be congratulated on their important work and the manner of its presentation.
Response:
Thank you for highlighting this important limitation. The smaller number of participants aged 40 years and older compared to those in the 18–29 age range reflects the demographic characteristics of our recruitment setting and methodology. Recruitment was conducted primarily within a university and through social media, which tend to have greater reach and engagement among younger adults [1]. Additionally, younger adults may have been more inclined to participate due to greater familiarity with and access to digital tools used in the study.
While we recognize the importance of a more balanced age distribution, logistical constraints and resource limitations made it challenging to actively target and recruit older participants. We have acknowledged this limitation in the manuscript and have added that future studies may benefit from ensuring a more balanced age distribution to explore food choice behaviors (lines 456-465).
- Leist, A. K. (2013). Social Media Use of Older Adults: A Mini-Review. Gerontology, 59(4), 378–384. https://doi.org/10.1159/000346818
Reviewer 3 Report
Comments and Suggestions for Authors
This manuscript uses data from 756 participants to examine the relationship between socioeconomic factors and food choice motives, to find that age, gender, studies and marital status make the biggest difference.
The justification is that there is scarce literature about food choices, but a quick search in WOS (food choice motives) renders almost 1500 articles, 100 per year in the last decade. Of them, >180 mention socioeconomic factors, 16 with the string "Latin America". Although a much more careful revision would be required, at first sight, there are plenty of previous works on the topic, so it would be necessary to justify better the novelty of the article, including the reasons (if any) that make the Mexican population special.
For the analysis, it would be useful to provide some global statistics on the overall performance of the model (fitness, predictive ability, etc.) and on the presence of collinearity. In line with this, in the discussion, the authors point to alternative explanations for the patterns in the data that could be disentangled if the interactions among factors were examined. Alternatively, PCA or cluster analyses could help find consistent profiles.
As for the conclusion, although the authors discuss how the findings of the study could inspire group-tailored measures, there is potential for further development of these ideas.
Author Response
Subject: Reply to comments of reviewers regarding the manuscript “Associations of sociodemographic characteristics with food choice motives’ importance among Mexican adults: a cross-sectional analysis”
Date: December 26th, 2024
To the reviewers of Foods
Manuscript Foods-3373288
Respected reviewers,
On behalf of the research team, we thank you for your valuable comments on our work. This review process is crucial to improve our manuscript.
In the next lines, you fill find a point-by-point response at each one of your questions/observations. It is important to notice that all references to specific lines modifications relate to the .pdf file, to avoid confusions regarding different Office layouts or configurations. We hope that with these improvements our manuscript is now enough suitable for publication.
Kind regards,
Corresponding authors.
-
Reviewer #3
- This manuscript uses data from 756 participants to examine the relationship between socioeconomic factors and food choice motives, to find that age, gender, studies and marital status make the biggest difference.
The justification is that there is scarce literature about food choices, but a quick search in WOS (food choice motives) renders almost 1500 articles, 100 per year in the last decade. Of them, >180 mention socioeconomic factors, 16 with the string "Latin America". Although a much more careful revision would be required, at first sight, there are plenty of previous works on the topic, so it would be necessary to justify better the novelty of the article, including the reasons (if any) that make the Mexican population special.
Thank you for your valuable observation. While searching on WOS and PubMed using the terms “food choices” or “food choice motives” yields several results, many of these studies actually focus on food consumption rather than on the importance of food choice motives. Additionally, the studies that do address the importance placed on food choice motives tend to concentrate on the socio-economic aspect as the sole sociodemographic factor. By performing a subsequent search, we were able to identify two studies in Latin America (in Brazil and Uruguay) relevant to our research, which have been added in lines 76-78. However, to our knowledge there are no similar studies in Mexico. Therefore, we have modified the introduction to clearly justify the relevance of this study (lines 75-84).
- For the analysis, it would be useful to provide some global statistics on the overall performance of the model (fitness, predictive ability, etc.) and on the presence of collinearity. In line with this, in the discussion, the authors point to alternative explanations for the patterns in the data that could be disentangled if the interactions among factors were examined. Alternatively, PCA or cluster analyses could help find consistent profiles.
We thank you for highlighting this. We agree that providing global statistics on the model's performance and examining collinearity are crucial for improving the robustness of our findings. Specifically, we have now reported R², adjusted R2, and root MSE. In addition, we used Ramsey regression error specification (RESET) test to detect omitted variables in the model (lines). We have also addressed the presence of collinearity by assessing Variance Inflation Factors (VIFs) and tolerance (1/VIF) to evaluate multicollinearity among predictors. These details have been added to the Methods (lines 204-206) and Results (Tables 3 and 4) sections of the manuscript for clarity. In addition, we added as a footnote in Tables 3 and 4 that for all regression models there was no multicollinearity and no omitted variables.
In the Discussion, we acknowledge the potential influence of interactions among the sociodemographic factors and food choice motives. As the reviewer suggests, examining interactions could help disentangle the underlying relationships between these variables. However, due to the current models’ design, we were unable to explore interactions in detail. We have noted this limitation in the manuscript and suggested that future research could benefit from exploring interactions more thoroughly (lines 478-481).
Regarding the suggestion of applying Principal Component Analysis (PCA) or cluster analysis to identify consistent profiles, we appreciate this valuable suggestion. While such analyses could indeed provide deeper insights into the structure of food choice motives, we have opted for a simpler approach to maintain the focus of our study. However, we agree that using PCA or clustering methods could be a promising avenue for future research and have included this recommendation in the Discussion section to guide future studies (lines 472-481).
- As for the conclusion, although the authors discuss how the findings of the study could inspire group-tailored measures, there is potential for further development of these ideas.
Thank you for your observation. We have expanded the potential uses of our results in lines 483-493, and summarized in them in the conclusion in lines 505-508.
Round 2
Reviewer 1 Report
Comments and Suggestions for Authors
The authors have revised the manuscript according to my suggestions.